Review

ecology, microbiology, molecular biology

fish, teleost, gut, microbiome, aquaculture, review

**Author for correspondence:**
William Bernard Perry
e-mail: w.perry@bangor.ac.uk

One contribution to a Special Feature 'Application of ecological and evolutionary theory to microbiome community dynamics across systems' guest edited by Dr James McDonald, Dr Britt Koskella, Professor Julian Marchesi.

# The role of the gut microbiome in sustainable teleost aquaculture

William Bernard Perry[1], Elle Lindsay[2], Christopher James Payne[3], Christopher Brodie[4,5] and Raminta Kazlauskaite[2]

[1]Molecular Ecology and Fisheries Genetics Laboratory, Bangor University, Bangor, Gwynedd LL57 2UW, UK
[2]Institute of Biodiversity, Animal Health & Comparative Medicine, University of Glasgow, Glasgow G12 8QQ, UK
[3]Institute of Aquaculture, University of Stirling, Stirling FK9 4LA, UK
[4]Ecosystems and Environment Research Centre, University of Salford, Salford M5 4WT, UK
[5]School of Biological and Environmental Sciences, Liverpool John Moores University, Liverpool L3 5UG, UK

  WBP, 0000-0001-9596-3333; CJP, 0000-0001-8313-2292

As the most diverse vertebrate group and a major component of a growing global aquaculture industry, teleosts continue to attract significant scientific attention. The growth in global aquaculture, driven by declines in wild stocks, has provided additional empirical demand, and thus opportunities, to explore teleost diversity. Among key developments is the recent growth in microbiome exploration, facilitated by advances in high-throughput sequencing technologies. Here, we consider studies on teleost gut microbiomes in the context of sustainable aquaculture, which we have discussed in four themes: diet, immunity, artificial selection and closed-loop systems. We demonstrate the influence aquaculture has had on gut microbiome research, while also providing a road map for the main deterministic forces that influence the gut microbiome, with topical applications to aquaculture. Functional significance is considered within an aquaculture context with reference to impacts on nutrition and immunity. Finally, we identify key knowledge gaps, both methodological and conceptual, and propose promising applications of gut microbiome manipulation to aquaculture, and future priorities in microbiome research. These include insect-based feeds, vaccination, mechanism of pro- and prebiotics, artificial selection on the holo-genome, in-water bacteriophages in recirculating aquaculture systems (RAS), physiochemical properties of water and dysbiosis as a biomarker.

## 1. Introduction

Since its conception in the 1980s describing soil ecology [1], the term microbiome has evolved into an intensely studied area of research. In recent decades, this area has begun expanding from an anthropocentric and medically dominated field, into a taxonomically broad field, examining research questions in non-model species, from trees [2] to frogs [3], and increasingly, fish. The diversification in microbiome studies has been driven by increased access to next generation sequencing (NGS), a tool that is not reliant upon culture-based techniques, which often require previous knowledge of target microbes.

Currently, gut bacterial communities have been assessed in over 145 species of teleosts from 111 genera, representing a diverse range of physiology and ecology (figure 1*a*), often with similarities in bacterial phyla composition between fish species, dominated by Bacteroidetes and Firmicutes [5,6]. Non-model taxa from an array of aquatic ecosystems have had their gut microbiomes sequenced using NGS, with studies extending beyond species identification, into hypothesis testing which was once only feasible in model systems. Examples of studies on non-model teleost gut microbiomes range from those demonstrating rapid gut microbiome restructuring after feeding in clownfish (*Premnas biaculeatus*) [7] to the effect of differing environmental conditions, such as dissolved oxygen content, on the gut microbial diversity of blind cave fish (*Astyanax mexicanus*) [8].

Proc. R. Soc. B 287: 20200184

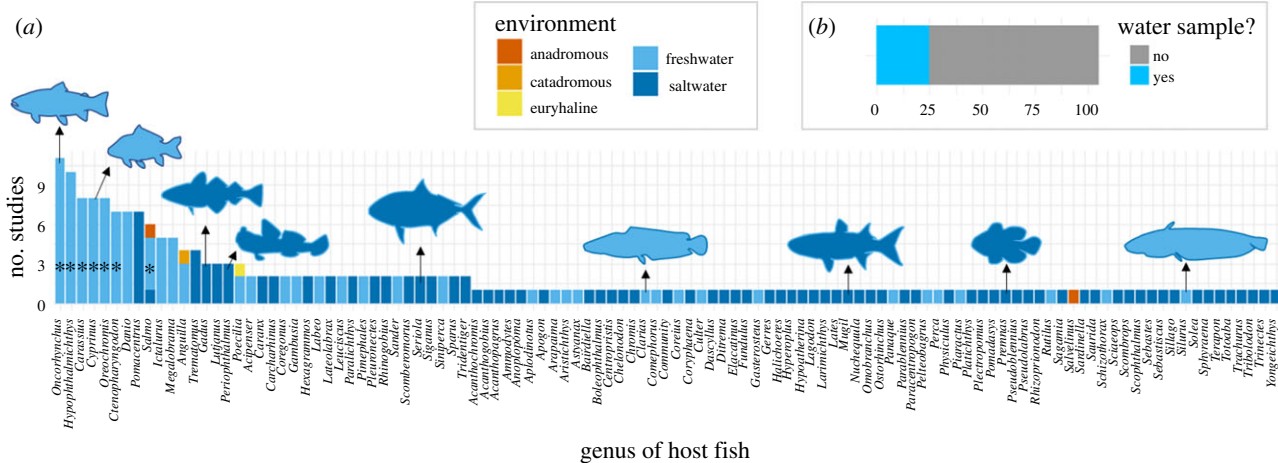

**Figure 1.** (a) Number of studies on the gut microbiome using NGS broken down by the genus of fish that the study was conducted on, as well as the environment those fish same from. Asterisk represents salmonid, carp and talapia. (b) The number of studies that assessed the water microbial communities. Gut microbiome studies were compiled using Web of Science [4] and only include studies that implemented NGS. It is acknowledged that total microbiome research extends further than this. Further information on search terms and filtering can be found in the electronic supplementary material. (Online version in colour.)

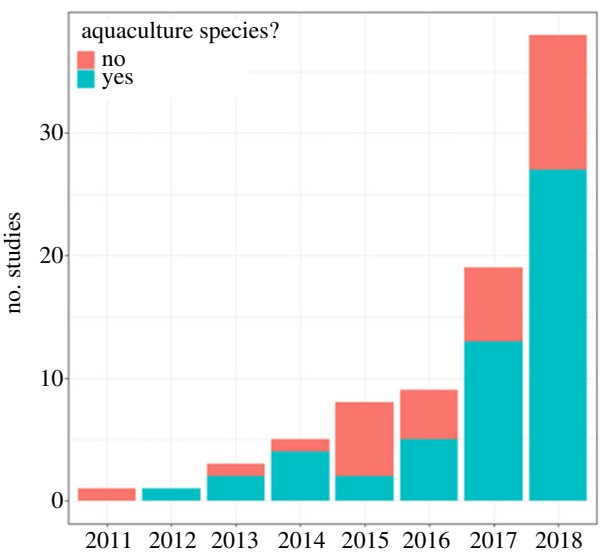

**Figure 2.** Growth in the studies using NGS on fish gut microbiomes, including food aquaculture species (aquaculture status taken from FishBase [12]). Further information on search terms and filtering can be found in the electronic supplementary material. (Online version in colour.)

Interest in the gut microbiome of fish has accelerated for many reasons, as not only do teleosts represent the most diverse vertebrate group [9], they are also of significant economic importance, including in aquaculture [10]. Aquaculture now provides over 45% of fish-based food products globally [11], and influence of the aquaculture industry on teleost gut microbiome research is demonstrated by the research questions tackled, with a clear bias towards salmonids (genera: *Oncorhynchus* and *Salmo*), carp (genera: *Hypophthalmichthys*, *Carassius*, *Cyprinus* and *Ctenopharyngodon*) and tilapia (genus: *Oreochromis*) (figure 2).

Rapid growth of the aquaculture industry has led to mounting pressure to make it more sustainable [13], and here we discuss four key components relevant to its sustainability in the context of the teleost gut microbiome: diet, immunity, artificial selection and closed-loop systems. We highlight some key deterministic factors important to aquaculture, although as shown in figure 3, there are numerous interacting

ecological processes. More in-depth reviews focusing on these specific interactions are available, for example, interactions between the gut microbiome and the immune system [14], energy homeostasis [15] and physiology [16]. Understanding and manipulating microbial–host–environmental interactions (figure 3a) and associated functional capacity in these areas could contribute substantially towards achieving a more sustainable aquaculture industry. We identify potential for future research, both methodological and conceptual. Other microbiomes are known to impact host function, in particular, the skin microbiome and its relationship to immunity [17], however, due to their differing ecology [18] and aquaculture applications [19], the gut microbiome will remain our focus here.

## 2. Diet

The gut microbiome has long been linked with diet, yielding insights into the commensal relationship between certain microbes and host. It has been shown that the teleost gut microbiome produces a range of enzymes (carbohydrases, cellulases, phosphatases, esterases, lipases and proteases) which contribute to digestion [10,20]. More intimate relationships also exist, for example, anaerobic bacteria in the teleost gut have a role in supplying the host with volatile fatty acids [21], an end product of anaerobic fermentation that provides energy for intestinal epithelial cells [22]. Gut microbes also synthesize vitamins and amino acids in the gut of aquatic vertebrates [23,24]. For example, the amount of vitamin $B_{12}$ positively correlated with the abundance of anaerobic bacteria belonging to the genera *Bacteroides* and *Clostridium*, in Nile tilapia (*Oreochromis niloticus*) [25]. Here, we discuss this host–microbe relationship in the context of contemporary aquaculture, with a focus on two timely issues: fishmeal and starvation.

### (a) Fishmeal

Fishmeal is an efficient energy source containing high-quality protein, as well as highly digestible essential amino and fatty acids [26], which is included in feed for a range of teleost species. Fish used in fishmeal production is, however,

**Figure 3.** (*a*) Schematic view of the deterministic processes that influence gut microbial communities in fish. Community assemblage of bacteria in the gut starts with inputs from the environment (green), such as the bacteria within the water column, or in solid particulates of biofilm, sediment and feed. Once ingested, these bacteria are influenced by interacting deterministic processes (brown) such as the host's abiotic gut environment, interaction with the hosts' physiology through the gut lining and its secretions, as well as interactions between other microbiomes. The outcome (red) is final community assembly, which can be characterized using an array of cutting-edge molecular techniques (purple). A subset of the boarder interactions is provided, with focus on (*b*) microbe–environment–host interactions, (*c*) host gut physiology and (*d*) behaviour. (Online version in colour.)

Proc. R. Soc. B **287**: 20200184

predominantly sourced from capture fisheries, putting pressure on already overfished stocks [13]. Despite a global decrease in fishmeal production, from an average of 6.0 million tonnes between 2001 and 2005 to 4.9 million tonnes between 2006 and 2010 [27], and growth in plant-based substitutes (e.g. wheat gluten, soya bean protein and pea protein), some aquaculture species still require a proportion of fish-sourced amino acids and proteins [28].

As dietary changes can alter the fish gut microbiome [29], there has been a considerable rise in the number of studies investigating the influence of alternative plant-protein sources on host–microbe interactions. Plant-protein sources have been shown to disturb the gut microbiota of some fish, with the production of antinutritional factors (factors that reduce the availability of nutrients) and antigens, impeding host resilience to stress [30], metabolism [31] and immune functioning [32]. Fish fed plant-protein-based diets can exhibit alterations in their intestinal morphology including disruption to the lamina propria and mucosal folds [33], which may modify attachment sites for commensal bacteria [34], and can therefore impact microbial composition [32,35].

Insect meal is increasingly used in aquafeed as a protein source with a high nutritional value [36], and several studies have demonstrated its potential use in manipulating the gut microbiome in fish [37,38]. As insects are chitin rich, these diets have been associated with prebiotic effects, through increased representation of beneficial commensal bacteria such as *Pseudomonas* sp. and *Lactobacillus* sp., which in turn improves performance and health in some fish [37]. Despite this, however, the beneficial effects of chitin are species specific, with Atlantic cod (*Gadus morhua*) and several cyprinid species demonstrating increased growth rates on diets with varying levels of chitin, whereas tilapia hybrids (*O. niloticus* × *O. aureus*) and rainbow trout (*Oncorhynchus mykiss*) both display decreased growth rates [39]. Chitin can therefore not be described as a probiotic for all species. The influence of insect meal on microbial-mediated functions also

remains underexplored, with little known about the extent to which species-specific responses to a chitin-rich diet are microbially mediated [40], offering scope for future research.

## (b) Starvation

Starvation is common in the production of valuable species such as salmon [41], sea bream [42], halibut [43] and cod [44], prior to handling, transportation and harvest, but is also used as a method to improve fillet quality. However, starvation is likely to have a substantial impact on host–microbe interactions (figure 3*b*). Gut microbial communities of the Asian seabass (*Lates calcarifer*), for example, shifted markedly in response to an 8-day starvation period, causing enrichment of the phylum Bacteroidetes, but a reduction of Betaproteobacteria, resulting in transcriptional changes in both host and microbial genes [45]. Perturbation to the gut microbiome could lead to the opening of niches for other commensal or even pathogenic bacteria [46], especially if this is combined with the compromised immune system of a stressed host [47] (figure 3*d*). Even if all fish are terminated shortly after starvation, gut microbial community changes before termination could cause long-term impacts to the microbial composition of water and biofilters in closed recirculating aquaculture systems (RAS). RAS systems will be discussed in greater detail later in this review.

## 3. Immunity

Gut microbial communities have strong links to immunity [48], which is pertinent in fish as they are in constant contact with water, a source of pathogenic and opportunistic commensal microbes [49]. In addition to this, fish cultured intensively are often stocked at high densities, allowing for easier transmission of microbes. Therefore, a microbially diverse gut microbiome in aquaculture is important to prevent unfavourable microbial colonization [50], and although the mechanisms are not fully

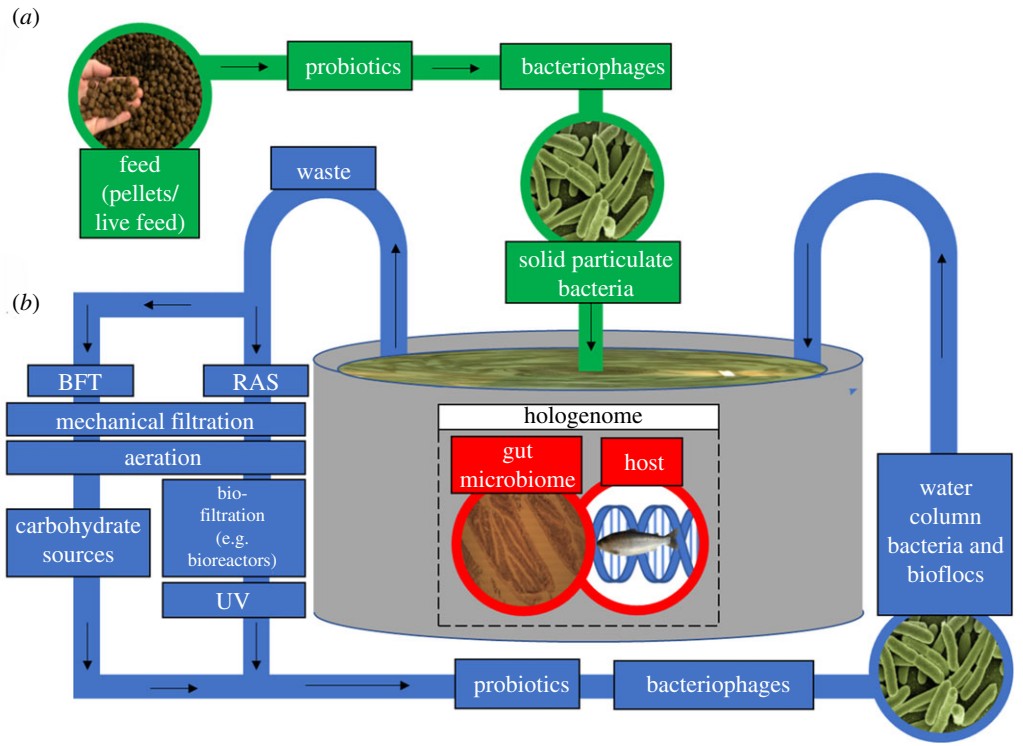

**Figure 4.** Schematic diagram of (*a*) feed inputs (green), (*b*) water processing (both RAS and BFT) (blue) and the (*c*) species being cultivated, along with its gut microbiome (red). (Online version in colour.)

understood, some key processes have been identified. For example, *Bacillus* and *Lactobacillus*, two common probiotic genera of bacteria used in aquaculture, are able to stimulate expression of inflammatory cytokines in the fish gut [51], increase the number of mucus layer producing goblet cells [52] and increase phagocytic activity [53]. Furthermore, comparison in gene expression between gnotobiotic zebrafish (*Danio rerio*) and conventionally reared zebrafish has shown bacteria induced expression of myeloperoxidase, an enzyme that allows neutrophil granulocytes to carry out antimicrobial activity [54]. Colonizing microbes can also modulate host gene expression to create favourable gut environments, thereby constraining invasion by pathogens [23], while also promoting expression of proinflammatory and antiviral mediators genes, leading to higher viral resistance [55]. Reducing viral and bacterial pathogens, such as *Vibrio* sp. and *Aeromonas* sp., is important for fish health in aquaculture, and will be discussed further in the context of closed-loop systems later in the review.

The interaction between the gut microbiome and the immune system is bilateral, for example, secretory immunoglobulins in fish recognize and coat intestinal bacteria to prevent them from invading the gut epithelium [56]. Similarly, in wild three-spined stickleback (*Gasterosteus aculeatus*), a causal chain (diet → immunity → microbiome) was discovered, demonstrating the impact of diet on fish immunity and thus the microbial composition of the gut [57]. Understanding microbial–host–environmental interactions like this are crucial for aquaculture, where, as previously discussed, diet is often manipulated.

## (a) Antibiotics

As most antibiotics used in aquaculture display broad-spectrum activity, they can affect both pathogens and non-target commensal microbes [58]. Oxytetracycline is one of the most widely used veterinary antibiotics, with 1500 metric

tonnes applied between 2000 and 2008 to salmon aquaculture in Chile [59]. However, oxytetracycline was seen to reduce gut microbial diversity in Atlantic salmon (*Salmo salar*), while enriching possible opportunistic pathogens belonging to the genus *Aeromonas*, and leading to a high prevalence of multiple tetracycline resistance-encoding bacterial genes [60]. Long-term exposure to oxytetracycline has also been reported to negatively affect growth, immunity and nutrient digestion/metabolism in Nile tilapia (*O. niloticus*) through antibiotic-induced disruption to the microbiota [61], causing considerable changes in the representation of Bacteroidetes and Firmicutes.

Vaccination has become a widespread prophylactic measure applied in aquaculture to improve immune functioning and disease resilience in farmed fish [62]. One study attempted to identify potential alterations in the microbiota structure and localized immune responses caused by a novel recombinant vaccine against *Aeromonas hydrophila* in grass carp (*Ctenopharyngodon idella*) [63]. Results from their study suggest that oral vaccines can target *Aeromonas* sp. through activation of innate and adaptive immune defences within the intestine without causing large disturbances in non-target microbiota populations. Given the importance of the immune response in regulating the gut microbiome [64], only a small number of studies have investigated the influence of vaccines on the resident microbiota composition and function in fish, providing grounds for future study.

## (b) Pro- and prebiotic supplementation

In view of the challenges associated with antibiotics, studies have examined the impact of alternative, prophylactic measures such as pro- and prebiotics (figure 4*a*). As literature on the types of pro- and prebiotics used in aquaculture have been reviewed elsewhere [65,66], as well as their effectiveness [67,68], we focus here on the ability of these compounds to induce changes in host physiology and function through shifts in the gut

microbiome. As has already been discussed, *Bacillus* sp. and *Lactobacillus* sp. have a beneficial effect on immunity and are suggested to provide an alternative approach to controlling disease in aquaculture. Targeted microbiota manipulation using these same bacteria have also been reported to exert beneficial effects on fish growth through (i) alterations in gut morphology [69], leading to improved digestion and metabolism [70] and (ii) microbial-mediated regulation of the genetic components involved in growth and appetite control [71,72]. Recently, the establishment of *Lactobacillus* probiotic bacteria within the gut microbiota was also associated with improved learning/memory capacity and changes in shoaling of zebrafish [73,74], indicating a potential gut–brain interaction pathway similar to what is described in higher vertebrates [75].

Research into the modulation of gut microbial communities using prebiotic compounds has expanded also. Certain dietary components have been reported to induce changes in gut morphology within the fish host, including vacuolation of enterocytes [76] and enhancing mucosal barrier integrity [77]. Improved mucosal protection and disease resilience are thought to be driven by microbes and associated microbial metabolites. Several prebiotics have been reported to manipulate the resident microbiota community of a host in favour of Firmicutes and short-chain fatty acid producing communities [78]. Mechanistic pathways remain elusive, however, with additional research required.

# 4. Artificial selection

Within aquaculture, selection has been applied routinely to increase production by enhancing desirable traits such as growth and disease resilience [79,80]. Recent evidence suggests, however, that host genetics plays a fundamental role in determining the gut microbiota in fish [81]. The 'hologenome' concept proposes that the host organism, along with their commensal microbial community, form one unit of selection [82]. Host physiology, for example, is determined in part by the host's genome and has the ability to shift gut microbiome composition, as demonstrated in zebrafish, whereby host neural activity and subsequent gut motility is able to destabilize microbial communities [46] (figure 3c). Although not described in teleosts, the reverse has also been seen, whereby microbial communities are able to regulate the host's gut through: (i) serotonin signalling [83,84], (ii) macrophages and enteric neurons interactions [85], (iii) metabolism of bile salts [86] and possibly, (iv) metabolism of short-chain fatty acids such as butyrate [87]. The host–microbe relationship means that traits selected during breeding programmes may be traits from the hologenome. Pyrosequencing studies have also shown significant changes in the microbial community composition of genetically improved fish compared with domesticated individuals [88,89]. Artificial selection has also been demonstrated on single species of bacteria, with *Aeromonas veronii* selected to exhibit greater colonization success in gnotobiotic zebrafish [90]. Environmental filtering of the reservoir of bacteria surrounding the fish generates the potential for improving colonization success of commensal bacteria. Currently, bacterial communities selected by breeding programmes could be neutral, sympathetic or antagonistic to the goals of artificial selection, and understanding this relationship will be vital in manipulating the hologenome.

# 5. Closed aquaculture systems

Many environmental problems plague current aquaculture practices. In addition to those already discussed, there are also issues with parasite transmission to wild fish [91], interactions between wild and escaped farmed fish [92], and release of faeces and excess feed into the environment [93]. One way to better control these problems is to remove aquaculture from ecosystems and bring it into a land-based setting [94].

## (a) Manipulating environmental microbiota

RAS and biofloc technology (BFT) are forms of aquaculture which use microbial communities to minimize excess nutrients and pathogens in rearing water (figure 4). In these systems, microbial reconditioning of the rearing water is vital as fish are stocked at high densities, resulting in elevated levels of organic material, which can promote microbial growth [95]. Selection of competitive, slow-growing K-strategist bacteria shifts the community from autotrophy to heterotrophy activity. Such shifts allow for a microbial community which maintains both water quality, through nutrient recycling, and inhibits the growth of fast-growing, opportunistic r-strategists, which include many bacterial pathogens such as *Aeromonas* sp. [96,97]. RAS and BFT could therefore be combined with vaccination against bacterial pathogens such as *Aeromonas* sp., as previously discussed, to reduce infections. The selection of K-strategist microbial communities differ between RAS and BFT. In RAS; K-selection is achieved by passing rearing water through heterotrophic biofilters [98], whereas in BFT, a high carbon to nitrogen ratio within rearing water is conditioned by the addition of carbohydrate sources, favouring heterotrophic K-strategist bacteria [99]. High-carbon conditions in BFT systems also promote nitrogen uptake into microbial biomass, which forms protein-rich bacterial 'flocs' that supplement feed [100].

Manipulation of microbes associated with live feed cultures is critical to the production of fish larvae as live feeds often contain opportunistic pathogens (figure 4a), resulting in stochastic mortality [64]. While traditional approaches involve non-selective, temporary methods (i.e. physical/chemical disinfection [101]), more recent efforts have shifted towards targeted manipulation through probiotics, for example, the successful use of *Phenylobacterium* sp., *Gluconobacter* sp. and *Paracoccus denitrificans* in rotifer (*Brachionus plicatilis*) production [102]. Lytic bacteriophages have also proven somewhat successful in reducing the prevalence of opportunistic pathogens, such as *Vibrio* sp. [103–105]. Live feed also appears to play a critical role in the delivery and establishment of colonizing gut microbiota in fish larvae upon first feeding [106]. Supplementation of live feed cultures with beneficial microbes, such as the previously mentioned *Lactobacillus* spp. and *Pediococcus* sp., has become common practice in hatcheries, with beneficial effects on growth, mucosal immunity and stress tolerance of larvae [17,107,108]. Bacteriophages and probiotics have also been applied directly to tank water (figure 4b); probiotics such as *Bacillus* spp. preventing fish mortality from *Vibrio* spp. infections [109] and *Flavobacterium columnare*-infecting phages have been shown to persist in RAS for up to 21 days [110]. Far less is known about the application of probiotics directly to tank water when compared with feed application [111]; however, and the use of bacteriophages is still in its infancy, providing potential for future research.

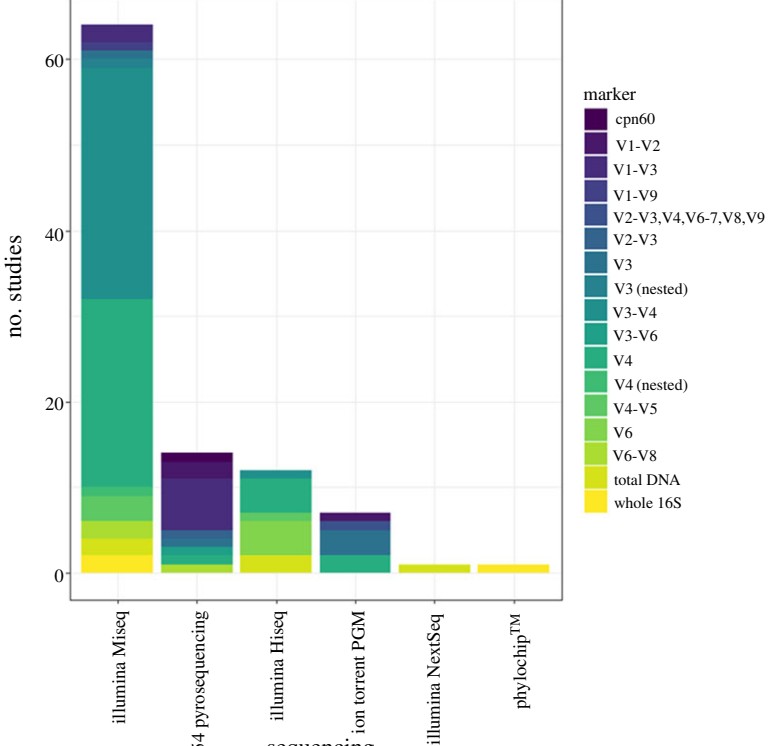

**Figure 5.** Methodological approaches used in high-throughput sequencing of fish gut microbiomes, broken down by the type of sequencing platform and genetic marker. Marker types are predominantly variable regions (V) within the 16S ribosomal RNA gene. Further information on search terms and filtering can be found in the electronic supplementary material. (Online version in colour.)

## (b) Controlling environmental variables

Changes in abiotic conditions in the water column propagate into the gut, as seen with dissolved oxygen concentration [8]. Such parameters are hard to control within the natural environment, but closed-loop systems provide consistent abiotic conditions, and allow for other variables, such as hologenome (figure 4c), to be manipulated with greater ease. The effect of many important physiochemical water properties (e.g. nitrate, ammonia and phosphate) on the teleost gut microbiome has not been studied, however, let alone how these properties interact [112]. Salinity is another important physiochemical property for the gut microbiome in many aquaculture species. When Atlantic salmon transition from freshwater to saltwater, individuals can experience a 100-fold increase in gut bacteria, combined with a shift in dominant microbial taxa [113]. Increasing salinity in RAS systems can, however, negatively impact nitrate removal in bioreactors [114], highlighting the importance of understanding interacting physiochemical properties.

## (c) Dysbiosis as a stress biomarker

The use of closed-loop systems is a progression to a more intensive method of aquaculture, mirroring the progression seen in animal agriculture, and a crucial element to sustainable intensification is welfare. It is possible to measure fish welfare through physiological and behavioural indicators, with a current focus on identifying stress. The microbiome has been identified as another potential biomarker [64] due to its interaction with the host immune system, and its responsive nature to stressors [115,116]. Therefore, identifying imbalances in the gut microbiome, or dysbiosis, could be a useful predictor of stress-related syndromes, which could ultimately lead to mortality. Using non-invasive faecal samples could

complement other non-invasive stress biomarkers, such as water cortisol [117], allowing for the optimization of husbandry, alerting operators to chemical (e.g. poor water quality, diet composition imbalance, accumulation of wastes), biological (e.g. overcrowding, social dominance, pathogens), physical (e.g. temperature, light, sounds, dissolved gases) or procedural (e.g. handling, transportation, grading, disease treatment) stressors [118]. More research is needed, however, in assessing the reliability and accuracy of faecal microbiome sampling in identifying stress.

## 6. Conclusion and future applications

The teleost gut microbiome has a clear role in the future of aquaculture, and although research has come a long way in recent decades, there are still many areas of gut microbiome research that require further development. As highlighted in figure 1b, there are still key elements lacking from many studies, particularly those assessing metacommunity composition, with the lack of water samples being particularly glaring. The ability to sample the environmental metacommunity with ease is one of the strengths of using a teleost model. Another methodological problem that will hinder comparability, reproducibility and metanalysis of fish gut microbiome datasets is the varying degree of sequencing platforms and markers (figure 5). A solution to this problem would be to focus on one marker, and one sequencing platform, with many metabarcoding microbiome studies adopting the V3 and V4 regions, sequenced on Illumina platforms. It is noted, however, that different markers and sequencing platforms work better in some systems with no simple fit-all approach. Therefore, tools that incorporate differences in taxonomic

identification that arise through using different methodological approaches will be vital in comparing datasets.

Current findings, as summarized here, show that the teleost gut microbiome plays an important role in aquaculture, however, the literature is dominated with studies performed on mammals, leading to limited data on functional capacity of fish gut microbiomes [64]. Furthermore, a knowledge gap exists between ascertaining the composition of the microbiome and understanding its function, partly due to the complexity and variability in the ecology of teleost gastrointestinal tracts [119] and unknown bacterial taxa. More specifically, however, it has been caused by the lack of synthesis between multiple cutting-edge molecular techniques. Progression in teleost gut microbiome research will depend on combining function (RNA sequencing), composition (metabarcoding and metagenomics) and spatial distribution (fluorescence *in situ* hybridization). Understanding host genetic diversity (population genomics) and expression (RNA sequencing) of that diversity, all while incorporating environmental variation, will also be vital.

Finally, there are many areas in which synergies between gut microbiomes and aquaculture can be made. These have been highlighted through the review, but, in summary, include a better understanding of the gut microbiome with respect to insect-based feeds, vaccination, mechanism of pro- and prebiotics, artificial selection on the hologenome, in-water bacteriophages in RAS/BFT, physiochemical properties of water and dysbiosis as a biomarker.

Data accessibility. All data collected in the systematic review can be found in the electronic supplementary material.

Authors' contributions. W.B.P. organized the creation of the review and facilitated communications between authors. W.B.P., E.L., R.K., C.J.P. and C.B. contributed to the concept and writing of the review.

Competing interests. No competing interests.

Funding. This study was funded by Natural Environment Research Council.

Acknowledgements. Thanks to Professor Gary Carvalho and Dr Martin Llewellyn for providing feedback on the review. This work was supported by the Natural Environment Research Council.

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
