## [Reviewer comments · Proceedings of the Royal Society B: Biological Sciences]

Review History

RSPB-2019-1681.R0 (Original submission)

Review form: Reviewer 1

Recommendation

Major revision is needed (please make suggestions in comments)

Scientific importance: Is the manuscript an original and important contribution to its field?

Marginal

General interest: Is the paper of sufficient general interest?

Marginal

Quality of the paper: Is the overall quality of the paper suitable?

Poor

Is the length of the paper justified?

No

Should the paper be seen by a specialist statistical reviewer?

No

Do you have any concerns about statistical analyses in this paper? If so, please specify them explicitly in your report.

No

It is a condition of publication that authors make their supporting data, code and materials available - either as supplementary material or hosted in an external repository. Please rate, if applicable, the supporting data on the following criteria.

Is it accessible?

N/A

Is it clear?

N/A

Is it adequate?

N/A

Do you have any ethical concerns with this paper?

No

Comments to the Author

This manuscript includes a summary of how ecological processes can impact the teleost microbiome and functional influences of the microbiome on the fish host, and then discusses how the fish microbiome may be manipulated in an aquaculture setting.

However, I think the authors have attempted to cover far too much. The manuscript lacks focus, detail or insightful discussion of how ecological and evolutionary processes shape the fish microbiome.

I think the manuscript would be better focusing specifically on aquaculture, and how a better understanding of ecological and evolutionary processes shaping aquaculture microbiomes could help improve its sustainability.

There is a lack of appreciation for the body of work elucidating mechanisms of microbiome community assembly in zebrafish models, or the influence of environmental/genetic factors in ecological models (stickleback/ guppy etc). It seems to me that there is a large gap between this mechanistic research, and microbiome research that has been carried out to date in most aquaculture-relevant species. I think it would be worthwhile to highlight this, and focus on discussing how application of community ecology may help us better understand aquaculture-relevant microbiomes.

Specific comments:

Abstract: there is reference to anthropogenic threats and environmental change, which are not mentioned anywhere in the main text.

The introduction/aim is far too general, and it is not clear to me what this article aims to bring to the field that is novel. I suggest a general introduction to the breadth of microbiome research in fish (including reference to existing specific reviews such as Butt & Volkoff 2019, Tarneki et al. 2017, Kelly & Salinas 2017, Burns & Guillemin 2017, Wang et al 2017). Then if aquaculture is to be the focus of the manuscript, it needs a better introduction on importance of improving its sustainability.

Figure 1 is informative, but it would be good to have a better description of fish species studied in the text, acknowledging the huge diversity in fish species (physiology and ecology), and bias in salmonid based research.

The authors say they aim to apply a community ecology perspective to the current understanding of teleost gut microbiomes, but this is not really the case. I find the discussion of this to be quite disjointed, and lacking depth or insight. For example, the final paragraph in section 2 seems to have been just added on at the end, and is not explored in a fish context. It would be much better to explore how these factors influence community assembly mechanisms throughout the discussion in an integrated way.

I agree that diet and environment are fundamental deterministic factors influencing the fish microbiome (section 2, para 2). But, this paragraph does not do justice to the large body of research examining this in fish. The references chosen seem quite random, and only represent a fraction of relevant research. Salinity is only one factor- many other environmental variables have been shown to be important. There is also no recognition of how environmental and host-specific deterministic factors will vary considerably within different fish species (huge diversity in physiology, ecology) or developmental stages.

There are existing comprehensive reviews of known functional influences of the fish microbiome that are not referred to (e.g. Butt and Volkoff 2019). Additionally the authors do not mention the limited number of true functional microbiome studies in aquaculture-relevant species, and difficulties with large number of unknown bacterial taxa, and limitations of accurate functional assignment etc.

I think Section 4 is the best part of the manuscript. I would suggest expanding this section to include how community ecology theory could be applied to better understand aquaculture microbiomes as has been done in other systems, and then highlighting how this knowledge could be applied to improve aquaculture sustainability in a holistic way. I think this should be the main focus of the manuscript.

There is no concluding section, or any perspective given on future research priorities etc.

Review form: Reviewer 2

Recommendation

Reject – article is not of sufficient interest (we will consider a transfer to another journal)

Scientific importance: Is the manuscript an original and important contribution to its field?

Marginal

General interest: Is the paper of sufficient general interest?

Marginal

Quality of the paper: Is the overall quality of the paper suitable?

Marginal

Is the length of the paper justified?

Yes

Should the paper be seen by a specialist statistical reviewer?

No

Do you have any concerns about statistical analyses in this paper? If so, please specify them explicitly in your report.

No

It is a condition of publication that authors make their supporting data, code and materials available - either as supplementary material or hosted in an external repository. Please rate, if applicable, the supporting data on the following criteria.

Is it accessible?

N/A

Is it clear?

N/A

Is it adequate?

N/A

Do you have any ethical concerns with this paper?

No

Comments to the Author

The title of this review article states that it will address microbial community assembly in the guts of teleosts in terms of assembly, function and significance. However, the article falls short of this objective. I did not learn too much of how communities are structured in terms of deterministic and stochastic processes. If there is not enough research to form a conclusion, then the authors can make this clear and provide a roadmap of what kinds of research are needed to address critical issues of the assembly and determinants of community structure.

The review is stronger when it comes to reviewing specific areas of interest such as the role of diet in community structure. But it seems obvious that diet, host genotype, antibiotics, etc. will cause a shift in community structure, but I am no closer in understanding how these data inform a broader, synthetic model of the determinants of community structure. Figure 3 is a start but falls short of what is typically published in the macro-community ecology literature. As one example, the authors state that neutral modeling has proven useful, but there is no indication of how and why it has been useful (p. 8). Also, how can intra- and inter-specific competition, predation, mutualism, etc. of microbial species be woven into a model of microbial community structure. Surely these interactions will be key in community structure, as they are in macroecology, but these are not addressed. In addition, should all OTUs be included in these considerations, or just "core" members? I am left wondering if it is possible to critically address community assembly in this system even though with "wild type" and germ-free fish and a tractable experimental system it seems that the determinants of community structure should be within reach.

The abstract is devoid of results and conclusions. Please provide some.

The paper ends abruptly without a synthesis and conclusions section. Again Fig. 3 is a start but leaves out key factors (e.g. environmental temperature, population density of hosts, competition) and seems overly simplistic. It would be helpful to present key experimental manipulations that can test current hypotheses of community assembly and determinants of community structure.

Review form: Reviewer 3

Recommendation

Major revision is needed (please make suggestions in comments)

Scientific importance: Is the manuscript an original and important contribution to its field?

Good

General interest: Is the paper of sufficient general interest?

Good

Quality of the paper: Is the overall quality of the paper suitable?

Good

Is the length of the paper justified?

Yes

Should the paper be seen by a specialist statistical reviewer?

No

Do you have any concerns about statistical analyses in this paper? If so, please specify them explicitly in your report.

No

It is a condition of publication that authors make their supporting data, code and materials available - either as supplementary material or hosted in an external repository. Please rate, if applicable, the supporting data on the following criteria.

Is it accessible?

N/A

Is it clear?

N/A

Is it adequate?

N/A

Do you have any ethical concerns with this paper?

No

Comments to the Author

See attached file. (See Appendix A)

Decision letter (RSPB-2019-1681.R0)

10-Oct-2019

Dear Mr Perry:

I am writing to inform you that your manuscript RSPB-2019-1681 entitled "Teleost microbiomes and aquaculture: assembly, function and significance" has, in its current form, been rejected for publication in Proceedings B.

This action has been taken on the advice of referees, who have recommended that substantial revisions are necessary. With this in mind we would be happy to consider a resubmission, provided the comments of the referees are fully addressed. However please note that this is not a provisional acceptance.

Sincerely,
Professor Hans Heesterbeek, Editor
mailto: proceedingsb@royalsociety.org

Comment by the Associate Editor:

Although all three reviewers express significant concerns. All three reviewers express concerns about the clarity, focus, significance and importance. They provided a very clear and unified message on what would be required to make the review of suitable broad interest and quality for Proc B. The manuscript in its current form is some way off that level. It would be important to address these comments in any revision of this manuscript, and particularly, to address the comments about a lack of coverage of key topics, clear results/insights (e.g. in terms of microbiome assembly, and deterministic vs stochastic processes, and methodological approaches, for example, are all highly active areas of microbiome science that would benefit from data synthesis from areas such as aquaculture, where a relatively larger body of research has been conducted) and a conclusion/synthesis to summarise key observations. The latter is lacking completely in the current version. Whether this can realise the potential to become a worthy contribution will depend on the degree to which the authors are able to engage and execute the suggestions made by the reviewers.

Reviewer(s)' Comments to Author:

Referee: 1

Comments to the Author(s)

This manuscript includes a summary of how ecological processes can impact the teleost microbiome and functional influences of the microbiome on the fish host, and then discusses how the fish microbiome may be manipulated in an aquaculture setting.

However, I think the authors have attempted to cover far too much. The manuscript lacks focus, detail or insightful discussion of how ecological and evolutionary processes shape the fish microbiome.

I think the manuscript would be better focusing specifically on aquaculture, and how a better

understanding of ecological and evolutionary processes shaping aquaculture microbiomes could help improve its sustainability.

There is a lack of appreciation for the body of work elucidating mechanisms of microbiome community assembly in zebrafish models, or the influence of environmental/genetic factors in ecological models (stickleback/ guppy etc). It seems to me that there is a large gap between this mechanistic research, and microbiome research that has been carried out to date in most aquaculture-relevant species. I think it would be worthwhile to highlight this, and focus on discussing how application of community ecology may help us better understand aquaculture-relevant microbiomes.

Specific comments:

Abstract: there is reference to anthropogenic threats and environmental change, which are not mentioned anywhere in the main text.

The introduction/aim is far too general, and it is not clear to me what this article aims to bring to the field that is novel. I suggest a general introduction to the breadth of microbiome research in fish (including reference to existing specific reviews such as Butt & Volkoff 2019, Tarneki et al. 2017, Kelly & Salinas 2017, Burns & Guillemin 2017, Wang et al 2017). Then if aquaculture is to be the focus of the manuscript, it needs a better introduction on importance of improving its sustainability.

Figure 1 is informative, but it would be good to have a better description of fish species studied in the text, acknowledging the huge diversity in fish species (physiology and ecology), and bias in salmonid based research.

The authors say they aim to apply a community ecology perspective to the current understanding of teleost gut microbiomes, but this is not really the case. I find the discussion of this to be quite disjointed, and lacking depth or insight. For example, the final paragraph in section 2 seems to have been just added on at the end, and is not explored in a fish context. It would be much better to explore how these factors influence community assembly mechanisms throughout the discussion in an integrated way.

I agree that diet and environment are fundamental deterministic factors influencing the fish microbiome (section 2, para 2). But, this paragraph does not do justice to the large body of research examining this in fish. The references chosen seem quite random, and only represent a fraction of relevant research. Salinity is only one factor- many other environmental variables have been shown to be important. There is also no recognition of how environmental and host-specific deterministic factors will vary considerably within different fish species (huge diversity in physiology, ecology) or developmental stages.

There are existing comprehensive reviews of known functional influences of the fish microbiome that are not referred to (e.g. Butt and Volkoff 2019). Additionally the authors do not mention the limited number of true functional microbiome studies in aquaculture-relevant species, and difficulties with large number of unknown bacterial taxa, and limitations of accurate functional assignment etc.

I think Section 4 is the best part of the manuscript. I would suggest expanding this section to include how community ecology theory could be applied to better understand aquaculture microbiomes as has been done in other systems, and then highlighting how this knowledge could be applied to improve aquaculture sustainability in a holistic way. I think this should be the main focus of the manuscript.

There is no concluding section, or any perspective given on future research priorities etc.

Referee: 2

Comments to the Author(s)

The title of this review article states that it will address microbial community assembly in the guts of teleosts in terms of assembly, function and significance. However, the article falls short of this objective. I did not learn too much of how communities are structured in terms of deterministic and stochastic processes. If there is not enough research to form a conclusion, then the authors can make this clear and provide a roadmap of what kinds of research are needed to address critical issues of the assembly and determinants of community structure.

The review is stronger when it comes to reviewing specific areas of interest such as the role of diet in community structure. But it seems obvious that diet, host genotype, antibiotics, etc. will cause a shift in community structure, but I am no closer in understanding how these data inform a broader, synthetic model of the determinants of community structure. Figure 3 is a start but falls short of what is typically published in the macro-community ecology literature. As one example, the authors state that neutral modeling has proven useful, but there is no indication of how and why it has been useful (p. 8). Also, how can intra- and inter-specific competition, predation, mutualism, etc. of microbial species be woven into a model of microbial community structure. Surely these interactions will be key in community structure, as they are in macro-ecology, but these are not addressed. In addition, should all OTUs be included in these considerations, or just "core" members? I am left wondering if it is possible to critically address community assembly in this system even though with "wild type" and germ-free fish and a tractable experimental system it seems that the determinants of community structure should be within reach.

The abstract is devoid of results and conclusions. Please provide some.

The paper ends abruptly without a synthesis and conclusions section. Again Fig. 3 is a start but leaves out key factors (e.g. environmental temperature, population density of hosts, competition) and seems overly simplistic. It would be helpful to present key experimental manipulations that can test current hypotheses of community assembly and determinants of community structure.

Referee: 3

Comments to the Author(s)

See attached file

Author's Response to Decision Letter for (RSPB-2019-1681.R0)

See Appendix B.

RSPB-2020-0184.R0

Review form: Reviewer 2

Recommendation

Accept with minor revision (please list in comments)

Scientific importance: Is the manuscript an original and important contribution to its field?
Good

General interest: Is the paper of sufficient general interest?
Acceptable

Quality of the paper: Is the overall quality of the paper suitable?
Acceptable

Is the length of the paper justified?
Yes

Should the paper be seen by a specialist statistical reviewer?
No

Do you have any concerns about statistical analyses in this paper? If so, please specify them explicitly in your report.
No

It is a condition of publication that authors make their supporting data, code and materials available - either as supplementary material or hosted in an external repository. Please rate, if applicable, the supporting data on the following criteria.

Is it accessible?
N/A

Is it clear?
N/A

Is it adequate?
N/A

Do you have any ethical concerns with this paper?
No

Comments to the Author

The authors have largely addressed the reviewers' comments in the revision.

The Immunity section was not related to aquaculture. I think this should be done since the objective of the revision is to review studies of the microbiome as related to aquaculture.

The Artificial Selection section includes the concept of the hologenome. This can be opening a can of worms. At least the role of the environmental reservoir of bacteria should be mentioned.

In section 5 (iii), The authors can make the connections between dysbiosis, welfare and rearing conditions more explicit. Presumably the idea is to change rearing conditions to minimize stress, maximize growth, etc.

In the Conclusions section, the authors could make it clearer exactly how filling in specific missing pieces of data will advance the efficacy of aquaculture. If it makes sense to do so, the authors can also prioritize the types of data to collect or to make it clear that it is necessary to collect some types of data before it makes sense to collect other types of data.

Figure 3 is not fully described in the text.

70: Can the authors tie specific interactions to specific references?

148: Cytokine production is a double-edge sword, but I take the authors' point.

150: The reference to bacteria-specific gene expression lacks context. What genes are expressed and to what end?

186: What is meant by "direct opposed response"?

222: What is meant by "proteins encoding for immune parameters"?

229: presumably genetic components of the host?

324: It seems that what are lacking are not elements of experimental design but lack of response variables.

340: Please define "mechanisms that underly [sic] function".

Carefully check the manuscript for typos and other errors, e.g., lines 92 ("examplem"), 139 ("all fish in a are") and 152 (micrboes).

Review form: Reviewer 3

Recommendation

Accept with minor revision (please list in comments)

Scientific importance: Is the manuscript an original and important contribution to its field?

Good

General interest: Is the paper of sufficient general interest?

Good

Quality of the paper: Is the overall quality of the paper suitable?

Good

Is the length of the paper justified?

Yes

Should the paper be seen by a specialist statistical reviewer?

No

Do you have any concerns about statistical analyses in this paper? If so, please specify them explicitly in your report.

No

It is a condition of publication that authors make their supporting data, code and materials available - either as supplementary material or hosted in an external repository. Please rate, if applicable, the supporting data on the following criteria.

Is it accessible?

Yes

Is it clear?

Yes

Is it adequate?

Yes

Do you have any ethical concerns with this paper?

No

Comments to the Author

Review of Perry et al ProcB Revision

This manuscript has had quite an overhaul since the previous round of review. I think it does a far better job of satisfying the brief / matching the title now. I find the figures to be improved, especially Fig. 3.

If I had one criticism of the revision it's that it still feels like a suite of disparate sections stuck together in one paper, rather than a contiguous synthesis of microbiome aquaculture relationships. For example multiple sections mention *Lactobacillus*, first as immune modulators and then as probiotics, but these sections don't flow into, or reference, one another. Secondly, the MS mentions RAS aquaculture in the first few paragraphs but doesn't then use it to reference the entire dedicated section on RAS at the end of the manuscript where more, vital detail can be found. This means it feels like these sections could have appeared in any order in the paper without affecting the structure, and also then sounds a bit repetitive, which seems jarring. I appreciate the authors have a lot of ground to cover here, and so I think my criticism is an emergent property of that. But I think there is still room for improvement here.

Below I list some issues where I think further clarity is needed as there are quite a few places throughout the MS where the language feels a bit imprecise. I also found a couple of typos.

Line 53: I think it would be good to expand the description of these examples to see how the gut microbiome is relevant. For example, is the clownfish example looking at the influence of feeding rate on the gut microbiome?

Line 92 "example". I'd also suggest you split the sentence here into two separate ones to avoid the excess of commas

Line 98 is there an 'and' missing after 'high quality protein'?

Line 111: I think you should state that these antinutrients are found in the soy meal as it's not clear what these relate to in the current sentence

Line 114: 'host function' is a bit vague here

Line 118: are there references for the statement about altered microbial diversity?

Line 122: is this because chitin is a prebiotic? Might be worth spelling out for a non-specialist reader

Line 141: I like this section on starvation, but could you expand on why starvation-mediated microbiome changes might be important in RAS aquaculture? I don't follow it.

Line 144: Too many commas in this sentence for my liking as I think It could flow better

Line 146: What's a robust gut microbiome? How do we measure it ?

Line 152 “microbes”

Line 161: Is the wild stickleback causal chain study missing a reference? I think it's this study Friberg IM, Taylor JD and Jackson JA (2019) Diet in the Driving Seat: Natural Diet-Immunity-Microbiome Interactions in Wild Fish. *Front. Immunol.* 10:243

Line 286: I know you reference fig 4b, but what are the promising results here?

Figure 3: Much improved from V1 of the manuscript, but I can't see the a-d subsections as indicated in the legend that I assume relate to the bottom right corner. Also where is 3c?

Decision letter (RSPB-2020-0184.R0)

21-Feb-2020

Dear Mr Perry:

Your manuscript has now been peer reviewed and the reviews have been assessed by an Associate Editor. The reviewers' comments (not including confidential comments to the Editor) and the comments from the Associate Editor are included at the end of this email for your reference. As you will see, the reviewers and the Associate Editor are positive, but have raised some issues with your manuscript that need to be addressed.

Research ethics:

Use of animals and field studies:

Please submit a copy of your revised paper within three weeks. If we do not hear from you within this time your manuscript will be rejected. If you are unable to meet this deadline please let us know as soon as possible, as we may be able to grant a short extension.

Best wishes,
Professor Hans Heesterbeek
<mailto:proceedingsb@royalsociety.org>

Associate Editor Board Member

Comments to Author:

I agree with the reviewers that the manuscript is much improved but requires further minor revisions. The reviewers highlight that further edits are needed to:

1. Better synthesise the disparate sections of the manuscript
2. Ensure that the immunity section is less general and more focussed on aquaculture.
3. Discuss the role of the environmental reservoir of bacteria in the section that considers artificial selection.
4. Make the connections between dysbiosis, welfare and rearing conditions more explicit.
5. Make it clearer exactly how filling in specific missing pieces of data will advance the efficacy of aquaculture.

On reading the manuscript, I also have some further suggestions:

Considering that the review is focussed on the gut microbiome of teleost fish, it is curious that other than sporadic references to specific taxa like *Lactobacillus* and *Bacteroides*, the manuscript does not provide a textual or graphical description of the taxonomic composition of the teleost gut microbiome, or the 'pathogens' that are referred to. Providing such information could assist the authors in addressing some of the points raised by the reviewers above, as they can specifically describe associations of specific taxa with diet, immunity, health and disease, and use this to illustrate some of the conceptual points made with empirical evidence. For example, which 'beneficial' taxa would be the best candidates as probiotics? And which deleterious microorganisms would bacteriophage applications be used to target? This in turn will be of benefit to the reader, who can use this information to interpret their own teleost microbiome datasets.

The authors address four key themes in the paper : 1) diet, 2) immunity, 3) artificial selection, 4) removing aquaculture from ecosystems (closed loop systems). But as they also numbered the sections in the manuscript, which means that themes 1, 2, 3 and 4, which follow the introduction, are numbered as sections 2, 3, 4 and 5 in the manuscript, which is a little jarring. I would suggest removing these section numbers to avoid this issue.

The manuscript needs a careful proofread for missing words, to correct variable spacing between words (e.g. after 'halibut' in line 129 and before 'immunity' in line 144), and typos.

Line 184 - the full genus name is needed for *A. hydrophila*

Reviewer(s)' Comments to Author:

Referee: 2

Comments to the Author(s).

The authors have largely addressed the reviewers' comments in the revision.

The Immunity section was not related to aquaculture. I think this should be done since the objective of the revision is to review studies of the microbiome as related to aquaculture.

The Artificial Selection section includes the concept of the hologenome. This can be opening a can of worms. At least the role of the environmental reservoir of bacteria should be mentioned.

In section 5 (iii), The authors can make the connections between dysbiosis, welfare and rearing conditions more explicit. Presumably the idea is to change rearing conditions to minimize stress, maximize growth, etc.

In the Conclusions section, the authors could make it clearer exactly how filling in specific missing pieces of data will advance the efficacy of aquaculture. If it makes sense to do so, the authors can also prioritize the types of data to collect or to make it clear that it is necessary to collect some types of data before it makes sense to collect other types of data.

Figure 3 is not fully described in the text.

70: Can the authors tie specific interactions to specific references?

148: Cytokine production is a double-edge sword, but I take the authors' point.

150: The reference to bacteria-specific gene expression lacks context. What genes are expressed and to what end?

186: What is meant by "direct opposed response"?

222: What is meant by "proteins encoding for immune parameters"?

229: presumably genetic components of the host?

324: It seems that what are lacking are not elements of experimental design but lack of response variables.

340: Please define "mechanisms that underly [sic] function".

Carefully check the manuscript for typos and other errors, e.g., lines 92 ("exaplem"), 139 ("all fish in a are") and 152 (micrboes).

Referee: 3

Comments to the Author(s).

Review of Perry et al ProcB Revision

This manuscript has had quite an overhaul since the previous round of review. I think it does a far better job of satisfying the brief / matching the title now. I find the figures to be improved, especially Fig. 3.

If I had one criticism of the revision it's that it still feels like a suite of disparate sections stuck together in one paper, rather than a contiguous synthesis of microbiome aquaculture relationships. For example multiple sections mention *Lactobacillus*, first as immune modulators and then as probiotics, but these sections don't flow into, or reference, one another. Secondly, the MS mentions RAS aquaculture in the first few paragraphs but doesn't then use it to reference the entire dedicated section on RAS at the end of the manuscript where more, vital detail can be found. This means it feels like these sections could have appeared in any order in the paper without affecting the structure, and also then sounds a bit repetitive, which seems jarring. I appreciate the authors have a lot of ground to cover here, and so I think my criticism is an emergent property of that. But I think there is still room for improvement here.

Below I list some issues where I think further clarity is needed as there are quite a few places throughout the MS where the language feels a bit imprecise. I also found a couple of typos.

Line 53: I think it would be good to expand the description of these examples to see how the gut microbiome is relevant. For example, is the clownfish example looking at the influence of feeding rate on the gut microbiome?

Line 92 "example". I'd also suggest you split the sentence here into two separate ones to avoid the excess of commas

Line 98 is there an 'and' missing after 'high quality protein'?

Line 111: I think you should state that these antinutrients are found in the soy meal as it's not clear what these relate to in the current sentence

Line 114: 'host function' is a bit vague here

Line 118: are there references for the statement about altered microbial diversity?

Line 122: is this because chitin is a prebiotic? Might be worth spelling out for a non-specialist reader

Line 141: I like this section on starvation, but could you expand on why starvation-mediated microbiome changes might be important in RAS aquaculture? I don't follow it.

Line 144: Too many commas in this sentence for my liking as I think It could flow better

Line 146: What's a robust gut microbiome? How do we measure it ?

Line 152 "microbes"

Line 161: Is the wild stickleback causal chain study missing a reference? I think it's this study Friberg IM, Taylor JD and Jackson JA (2019) Diet in the Driving Seat: Natural Diet-Immunity-Microbiome Interactions in Wild Fish. Front. Immunol. 10:243

Line 286: I know you reference fig 4b, but what are the promising results here?

Figure 3: Much improved from V1 of the manuscript, but I can't see the a-d subsections as indicated in the legend that I assume relate to the bottom right corner. Also where is 3c?

Author's Response to Decision Letter for (RSPB-2020-0184.R0)

See Appendix C.

Decision letter (RSPB-2020-0184.R1)

03-Apr-2020

Dear Mr Perry

I am pleased to inform you that your manuscript RSPB-2020-0184.R1 entitled "The role of the gut microbiome in sustainable teleost aquaculture" has been accepted for publication in Proceedings B.

The Associate Editor has recommended publication, but also suggests some minor revisions to your manuscript. Therefore, I invite you to respond to the comments and revise your manuscript. Because the schedule for publication is very tight, it is a condition of publication that you submit the revised version of your manuscript within 7 days. If you do not think you will be able to meet this date please let us know.

If you wish to submit your data to Dryad (<http://datadryad.org/>) and have not already done so you can submit your data via this link [http://datadryad.org/submit?journalID=RSPB&manu=\(Document not available\)](http://datadryad.org/submit?journalID=RSPB&manu=(Document%20not%20available)) which will take you to your unique entry in the Dryad repository. If you have already submitted your data to dryad you can make any necessary revisions to your dataset by following the above link. Please see <https://royalsocietypublishing.org/journal/rsos/ethics-policies/data-sharing-mining/> for more details.

6) For more information on our Licence to Publish, Open Access, Cover images and Media summaries, please visit <https://royalsocietypublishing.org/journal/rsos/authors/author-guidelines/>.

Sincerely,
 Professor Hans Heesterbeek
 Editor, Proceedings B
<mailto:proceedingsb@royalsocietypublishing.org>

Associate Editor:

Board Member

Comments to Author:

Thank you for the revised manuscript which has broadly addressed the reviewers and AE comments. There are some further minor revisions to the revised sections that need to be completed before the manuscript can be accepted.

Line 150 - 'For example, Bacillus and Lactobacillus, two of the most commonly used genus ' genus (singular) should be changed to 'genera' (plural).

Also here: 'As has already been discussed, probiotic bacteria belonging to the genus Bacillus and Lactobacillus'

And here: 'beneficial commensal bacteria, genus such as Pseudomonas and Lactobacillus, which in turn,'

Lines 155 and 267 - 'Vibro' should be Vibrio?

Line 226 - 228 - the term 'Artificial selection on the environmental reservoir of bacteria surrounding the fish' should be revised as 'artificial selection' cannot act directly upon the environmental reservoir of bacteria. Maybe 'environmental filtering' is a more appropriate term.

"Supplementation of live feed cultures with beneficial microbes, such as the previously mentioned Lactobacillus sp.," One species? If more than one species, need to edit to spp.. Also here; 'Bacillus sp. preventing loss from Vibrio sp. '

There seems to be a word missing in this sentence: 'and Flavobacterium column are -infecting phages'

Author's Response to Decision Letter for (RSPB-2020-0184.R1)

See Appendix D.

Decision letter (RSPB-2020-0184.R2)

06-Apr-2020

Dear Mr Perry

I am pleased to inform you that your manuscript entitled "The role of the gut microbiome in sustainable teleost aquaculture" has been accepted for publication in Proceedings B.

Your article has been estimated as being 10/10.5 pages long. Our Production Office will be able to confirm the exact length at proof stage.

Open Access

Paper charges

Sincerely,

Proceedings B

Appendix A

Review of Perry et al ProcB 2019

Perry et al present a review of the factors shaping the compositional and functional traits of host-associated bacterial communities in teleost fish, with a particular focus on the relevance for aquaculture.

Overall the paper is well written, well structured, and interesting. In a few parts, particularly in the first half, there are examples of unclear language that I think need tightening up. I have included details below. I also note the following two areas where I feel the manuscript could be improved.

- In several parts the paper draws attention to the issue that factors shaping the structure / composition of microbiomes is progressing, the effects on function are less well known. I think there's an opportunity here to make this issue a lot more prominent and state where the biggest knowledge gaps are. However this will require some text in the introduction that discusses the issues with mapping structure to function in microbial communities. I think what's missing from the MS as it stands is a synthesis that draws together all the previous sections. Much like comment 1 above, I think there's an opportunity for this paper to set itself apart from other 'things that affect the microbiome' reviews by identifying the major knowledge gaps and priorities for future research, and perhaps how these things might differ depending on the problem e.g. modulating immunity, modulating growth + improving yield.
- I found the lack of focus on the skin microbiome quite jarring and a real shame to exclude it from the review entirely, as stated in Section 1.4. The authors themselves acknowledge how important it is for immunity, and talk about how fish are directly exposed to the environment, so some discussion of host-skin microbiome-pathogen relationships would really strengthen the manuscript. There's obviously a lot of literature about how the environment shapes the skin microbiome in fish too, which you could include for comparison in the relevant sections on gut microbiomes, but I think the pathogen immunity section is the most vital here.

The figures are clear and informative. I especially like the infographic style of figure 1. I have some comments on the style of figure 3 that I include below.

Apologies for the vague section references- but there were no line numbers on the reviewing pdf. Quite frustrating for all.

Specific Comments

Introduction: though the first few sentences attempt to summarise the field of microbiome research, but this section is a little vague. For example, it does little to discuss decades of soil

microbiome research, especially that in the context of plant-microbe symbioses. Sure, you're focussing on bacterial microbiomes, but scientists have been culturing soil & environmental microbiomes for decades.

On a wider note, microbiome research, bacterial or otherwise, doesn't have to use NGS. I know you mention this below in paragraph 3, but it seems odd to do so after introducing the focus of the manuscript on fish. I would move paragraph 3 up to the 'general' part of the intro and then lead into the fish-specific text

Intro Paragraph 2: I think your justification for why teleost gut microbiomes possess ideal properties for study. For example 'non-maternal control' is a bit of a vague term, and there are many species for which the environment is a strong driver of microbiome structure (e.g. amphibians), or where traits like social environment have a large influence (mammals) - so you need to make a stronger case here. Also why is a 'broad array of host genetic, physiological and behavioural diversity' beneficial? This needs explaining. I would also expect to see some text here about things like the availability of genomic resources as being useful for this system. Update: I see you've addressed some of this in section 2 where you expand upon these traits, so perhaps the redundancy across these sections needs addressing.

Intro Paragraph 3 " (igure 2) "

Also Intro Para3: "provided evidence for the selective-nature of certain host-microbe interactions". Two things here - it's not clear what you mean here but I assume you mean host-microbe associations represent a non-random assembly from what's 'available' in the environment / selection by the host (or microbe) for specific partnerships. This should be more clearly stated. Secondly, there's plenty of evidence for this in amphibians and humans too, so the narrow taxonomic focus is a little unnecessary.

Paragraph 4: "The impact of diet as a deterministic force" is a little vague. Do you mean on microbiome structure and function?. Again, this paragraph is very narrowly focussed - why not draw more widely on the literature for things like gut-microbiota-brain axis research, microbe-immunity research etc? You can still draw attention to the wider literature before leading the reader to why understanding these interactions is vital for aquaculture science.

Para5: Can you have a [functional] role "on" physiology? I think "in" is more appropriate

Section 2 Para1: Much more clear; big fan of the overview given here.

2.1 : examples of environmental conditions here would be good as you switch from this sentence back into talking about diet/starvation

2.3: OTU not defined

2.4 “Host selective pressures” is a bit vague - better to spend a bit of space saying exactly what is meant here.

3.3 Do you mean ‘implicated’ strongly in immunity?

3.3: I don’t follow the logic here. All animals are in constant contact with their immediate environments. Also the logic of the importance of the gut microbiome in this context isn’t clear. What about the skin microbiome and pathogens that colonise the skin? That seems far more relevant when talking about direct environmental contact.

The *Lactobacillus* example here needs more detail beyond simply talking about ‘the immune response’. Which arm of the vertebrate immune response? And what pathogen does this help to protect against?

4.1 I like this section but it needs more detail given that it is about manipulation and aquaculture. Is it sufficient to talk about genetic selection changing the composition of microbiomes? Don’t we care more about microbiome function, how that is impacted by selection on host genes, and what that means for fish health? If there’s no evidence of functional changes, is this an opportunity to highlight the paucity of data? You do this well with the diet section below in reference to the unknown effect of insect protein on gut microbiome function.

4.3 Perhaps a reference to support the statement about disease outbreak and a reliance on chemotherapeutics?

4.4: You mention probiotics in section 4.3, which could of course be applied indirectly through the environment, so should probiotics alternatively be discussed in this section?

4.4: “allow for a microbial...” something missing here

Figure 3: I like what you’re trying to do here but I think it could be clearer. For example, section (a) should really include the effect of feed as prebiotics on the microbiome, but seems to only deal with direct microbial input. Section b seems odd when presented as 3 overlapping discs. I’d suggest it would be more informative to open this up to allow illustration of how the three discs are inter-related using arrows. For example, the abiotic gut env. Can affect both microbe-microbe and host-microbe interactions. Finally, section c is an opportunity to illustrate how we can use high throughput techniques to measure both the structural and functional characteristics of teleost microbiomes, but critically that these two traits a) tell us different things and b) do not always map 1:1 because of horizontal gene transfer, functional redundancy across microbial species, and the importance of understanding gene expression and not just gene presence in microbial genomes.

Appendix B

Response to reviewers' comments

Thank you all so much for your time and input, please find point by point responses below.

Comment by the Associate Editor:

Although all three reviewers express significant concerns. All three reviewers express concerns about the clarity, focus, significance and importance. They provided a very clear and unified message on what would be required to make the review of suitable broad interest and quality for Proc B. The manuscript in its current form is some way off that level. It would be important to address these comments in any revision of this manuscript, and particularly, to address the comments about a lack of coverage of key topics, clear results/insights (e.g. in terms of microbiome assembly, and deterministic vs stochastic processes, and methodological approaches, for example, are all highly active areas of microbiome science that would benefit from data synthesis from areas such as aquaculture, where a relatively larger body of research has been conducted) and a conclusion/synthesis to summarise key observations. The latter is lacking completely in the current version. Whether this can realize the potential to become a worthy contribution will depend on the degree to which the authors are able to engage and execute the suggestions made by the reviewers.

We thank you for your insightful comments, and we hope that the changes made here in the focus and structure of the review have addressed the problems with lack of real insight. We have also provided a conclusion and synthesis section for what we believe to be promising areas in gut microbiome research and aquaculture.

Reviewer(s)' Comments to Author:

Referee: 1

Comments to the Author(s)

This manuscript includes a summary of how ecological processes can impact the teleost microbiome and functional influences of the microbiome on the fish host, and then discusses how the fish microbiome may be manipulated in an aquaculture setting. However, I think the authors have attempted to cover far too much. The manuscript lacks focus, detail or insightful discussion of how ecological and evolutionary processes shape the fish microbiome.

We agree.

I think the manuscript would be better focusing specifically on aquaculture, and how a better understanding of ecological and evolutionary processes shaping aquaculture microbiomes could help improve its sustainability.

We very much agree with this suggestion, and have used it to restructure the entire review.

There is a lack of appreciation for the body of work elucidating mechanisms of microbiome community assembly in zebrafish models, or the influence of environmental/genetic factors in ecological models (stickleback/ guppy etc). It seems to me that there is a large gap between this mechanistic research, and microbiome research that has been carried out to date in most aquaculture-relevant species. I think it would be worthwhile to highlight this, and focus on discussing how application of community ecology may help us better understand aquaculture-relevant microbiomes.

Hopefully we have achieved this with the new restructured format. Thank you for your bold constructive feedback!

Specific comments:

Abstract: there is reference to anthropogenic threats and environmental change, which are not mentioned anywhere in the main text.

This has been removed, as it was not mentioned anywhere else in the review.

The introduction/aim is far too general, and it is not clear to me what this article aims to bring to the field that is novel.

Based on these comments, we have re-evaluated the focus, using sustainable aquaculture as a theme that runs throughout the review, hopefully making the narrative more coherent.

I suggest a general introduction to the breadth of microbiome research in fish (including reference to existing specific reviews such as Butt & Volkoff 2019, Tarneki et al. 2017, Kelly & Salinas 2017, Burns & Guillemin 2017, Wang et al 2017).

Comment to previous reviews added.

Then if aquaculture is to be the focus of the manuscript, it needs a better introduction on importance of improving its sustainability.

Addition of the sustainability of aquaculture and the role of the gut microbiome in 1) diet, 2) immunity, 3) artificial selection, 4) removing aquaculture from ecosystems (closed loop systems).

Figure 1 is informative, but it would be good to have a better description of fish species studied in the text, acknowledging the huge diversity in fish species (physiology and ecology), and bias in salmonid based research.

Comment on the breadth of fish species studied added into the introduction: "Examples of studies on non-model teleost gut microbiome range from those examining feeding rate in clownfish (Parris, Morgan and Stewart, 2019) to dissolved oxygen content in blind cave fish (Ornelas-García et al., 2018).".

Comment on salmonid biased added: "with a clear bias towards salmonids (genus:

Oncorhynchus and Salmo) carp (genus: Hypophthalmichthys, Carassius, Cyprinus and Ctenopharyngodon) and tilapia (genus: Oreochromis) (figure 1).”

The authors say they aim to apply a community ecology perspective to the current understanding of teleost gut microbiomes, but this is not really the case. I find the discussion of this to be quite disjointed, and lacking depth or insight. For example, the final paragraph in section 2 seems to have been just added on at the end, and is not explored in a fish context. It would be much better to explore how these factors influence community assembly mechanisms throughout the discussion in an integrated way.

We have removed the community ecology perspective, as we agree that it was not the strong point of the review, and was lacking insight. Instead we focus on aquaculture which we believe is the strongest part of the review.

I agree that diet and environment are fundamental deterministic factors influencing the fish microbiome (section 2, para 2). But, this paragraph does not do justice to the large body of research examining this in fish. The references chosen seem quite random, and only represent a fraction of relevant research. Salinity is only one factor- many other environmental variables have been shown to be important. There is also no recognition of how environmental and host-specific deterministic factors will vary considerably within different fish species (huge diversity in physiology, ecology) or developmental stages.

This has been removed from the introduction, as you are right, it was a random collection of examples.

There are existing comprehensive reviews of known functional influences of the fish microbiome that are not referred to (e.g. Butt and Volkoff 2019). Additionally the authors do not mention the limited number of true functional microbiome studies in aquaculture-relevant species, and difficulties with large number of unknown bacterial taxa, and limitations of accurate functional assignment etc.

Butt and Volkoff 2019 reference added, and the addition of functional assignment limitations: “Understanding and manipulating the teleost gut microbiome, microbial-host-environmental interactions (figure 3a) and their functional capacity in these areas could provide a substantial contribution in achieving a more sustainable aquaculture industry.”

I think Section 4 is the best part of the manuscript. I would suggest expanding this section to include how community ecology theory could be applied to better understand aquaculture microbiomes as has been done in other systems, and then highlighting how this knowledge could be applied to improve aquaculture sustainability in a holistic way. I think this should be the main focus of the manuscript.

We have reshaped the manuscript based on this suggestion, as we agree it is here the most insight was.

There is no concluding section, or any perspective given on future research priorities etc.

Concluding section added.

Referee: 2

Comments to the Author(s)

The title of this review article states that it will address microbial community assembly in the guts of teleosts in terms of assembly, function and significance. However, the article falls short of this objective. I did not learn too much of how communities are structured in terms of deterministic and stochastic processes. If there is not enough research to form a conclusion, then the authors can make this clear and provide a roadmap of what kinds of research are needed to address critical issues of the assembly and determinants of community structure.

Focus on community assembly has been removed, and the new focus of the review is aquaculture, while incorporating aspects of community assembly.

The review is stronger when it comes to reviewing specific areas of interest such as the role of diet in community structure. But it seems obvious that diet, host genotype, antibiotics, etc. will cause a shift in community structure, but I am no closer in understanding how these data inform a broader, synthetic model of the determinants of community structure.

New focus of the review on how these aspects of diet, genotype and antibiotics impact the microbiome through an aquaculture perspective, while incorporating understanding on community structure.

Figure 3 is a start but falls short of what is typically published in the macro-community ecology literature. As one example, the authors state that neutral modeling has proven useful, but there is no indication of how and why it has been useful (p. 8). Also, how can intra- and inter-specific competition, predation, mutualism, etc. of microbial species be woven into a model of microbial community structure. Surely these interactions will be key in community structure, as they are in macro-ecology, but these are not addressed. In addition, should all OTUs be included in these considerations, or just “core” members? I am left wondering if it is possible to critically address community assembly in this system even though with “wild type” and germ-free fish and a tractable experimental system it seems that the determinants of community structure should be within reach.

New focus of the review, which is no longer trying to identify the broad processes shaping community structure.

The abstract is devoid of results and conclusions. Please provide some.

Provided.

The paper ends abruptly without a synthesis and conclusions section.

Synthesis and conclusions section added.

Again Fig. 3 is a start but leaves out key factors (e.g. environmental temperature, population density of hosts, competition) and seems overly simplistic. It would be helpful to present key experimental manipulations that can test current hypotheses of community assembly and determinants of community structure.

Figure 3 has been revisited, with greater focus on interactions, while also complimenting it with experimental manipulations conducted in previous research, within the text of the review.

We hope that this new focus will provide greater insight – thank you for your constructive comments.

Referee: 3

Perry et al present a review of the factors shaping the compositional and functional traits of host-associated bacterial communities in teleost fish, with a particular focus on the relevance for aquaculture.

Overall the paper is well written, well structured, and interesting.

Thank you.

In a few parts, particularly in the first half, there are examples of unclear language that I think need tightening up. I have included details below. I also note the following two areas where I feel the manuscript could be improved.

- In several parts the paper draws attention to the issue that factors shaping the structure / composition of microbiomes is progressing, the effects on function are less well known. I think there's an opportunity here to make this issue a lot more prominent and state where the biggest knowledge gaps are. However this will require some text in the introduction that discusses the issues with mapping structure to function in microbial communities.

We have made a real effort to try and highlight knowledge gaps in this iteration of the review, based on these comments, and hopefully this shows.

I think what's missing from the MS as it stands is a synthesis that draws together all the previous sections. Much like comment 1 above, I think there's an opportunity for this paper to set itself apart from other 'things that affect the microbiome' reviews by identifying the major knowledge gaps and priorities for future research, and perhaps how these things might differ depending on the problem e.g. modulating immunity, modulating growth + improving yield.

With the new focus on aquaculture, we feel as if we have managed to draw the sections together in a far more coherent manner, which culminates, in what we think, is a useful synthesis/conclusion section.

- I found the lack of focus on the skin microbiome quite jarring and a real shame to exclude it from the review entirely, as stated in Section 1.4. The authors themselves acknowledge how important it is for immunity, and talk about how fish are directly exposed to the

environment, so some discussion of host-skin microbiome-pathogen relationships would really strengthen the manuscript. There's obviously a lot of literature about how the environment shapes the skin microbiome in fish too, which you could include for comparison in the relevant sections on gut microbiomes, but I think the pathogen immunity section is the most vital here.

As you said, we do agree that the skin microbiome is important, however, for brevity (as we are close to the maximum page number), and so not to lose focus, we have excluded it from the review, as we feel it contains enough literature for a review of its own. We also believe that because the skin microbiome is likely to have a different ecology to that of the gut, it will result in differing aquaculture applications, which would be too extensive to discuss in full here.

The figures are clear and informative.

Thank you!

I especially like the infographic style of figure 1. I have some comments on the style of figure 3 that I include below.

Apologies for the vague section references- but there were no line numbers on the reviewing pdf. Quite frustrating for all.

Apologies for this.

Specific Comments

Introduction: though the first few sentences attempt to summarise the field of microbiome research, but this section is a little vague. For example, it does little to discuss decades of soil microbiome research, especially that in the context of plant-microbe symbioses. Sure, you're focussing on bacterial microbiomes, but scientists have been culturing soil & environmental microbiomes for decades.

Addition of: "Since its conception in the 1980s describing soil ecology (Whipp, Lewis and Cooke, 1987) the term microbiome has evolved and grown into an intensely studied area of research" in the first line.

On a wider note, microbiome research, bacterial or otherwise, doesn't have to use NGS. I know you mention this below in paragraph 3, but it seems odd to do so after introducing the focus of the manuscript on fish. I would move paragraph 3 up to the 'general' part of the intro and then lead into the fish-specific text

We have tried to broaden the microbiome research included to not only include NGS, as we agree, there are lots of bacterial manipulation experiments that do not use NGS.

Intro Paragraph 2: I think your justification for why teleost gut microbiomes possess ideal properties for study. For example 'non-maternal control' is a bit of a vague term, and there are many species for which the environment is a strong driver of microbiome structure (e.g. amphibians), or where traits like social environment have a large influence (mammals) - so you need to make a stronger case here. Also why is a 'broad array of host genetic, physiological and behavioural diversity' beneficial? This needs explaining. I would also expect to see some text here about things like the availability of genomic resources as being useful for this system.

We have removed this, in fitting with the new aquaculture focus.

Update: I see you've addressed some of this in section 2 where you expand upon these traits, so perhaps the redundancy across these sections needs addressing.

Intro Paragraph 3 " (figure 2) "

Fixed.

Also Intro Para3: "provided evidence for the selective-nature of certain host-microbe interactions". Two things here - it's not clear what you mean here but I assume you mean host-microbe associations represent a non-random assembly from what's 'available' in the environment / selection by the host (or microbe) for specific partnerships. This should be more clearly stated. Secondly, there's plenty of evidence for this in amphibians and humans too, so the narrow taxonomic focus is a little unnecessary.

Removed.

Paragraph 4: "The impact of diet as a deterministic force" is a little vague. Do you mean on microbiome structure and function?. Again, this paragraph is very narrowly focused - why not draw more widely on the literature for things like gut-microbiota-brain axis research, microbe-immunity research etc? You can still draw attention to the wider literature before leading the reader to why understanding these interactions is vital for aquaculture science.

Restructured.

Para5: Can you have a [functional] role "on" physiology? I think "in" is more appropriate

Removed.

Section 2

Para1: Much more clear; big fan of the overview given here.

2.1 : examples of environmental conditions here would be good as you switch from this sentence back into talking about diet/starvation

2.3: OTU not defined

Removed, but defined later on.

2.4 "Host selective pressures" is a bit vague - better to spend a bit of space saying exactly what is meant here.

Agreed – discussion of host selective pressures is now incorporated in the artificial selection section, hopefully less vaguely.

3.3 Do you mean 'implicated' strongly in immunity?

Removed.

3.3: I don't follow the logic here. All animals are in constant contact with their immediate environments. Also the logic of the importance of the gut microbiome in this context isn't clear.

Yes, this is a good point. What we were trying to explain here was that the water is going to be full of bacteria (which of course the air is to, but to a lesser extent). The argument can be made that this is also true for amphibians. We have rephrased this to: “they are in constant contact with water, a source of pathogenic and opportunistic commensal microbes (Ellis, 2001)” at the start of the immunity section.

What about the skin microbiome and pathogens that colonise the skin? That seems far more relevant when talking about direct environmental contact.

Again, while I do agree with this wholeheartedly, I believe the constraints on length do not allow us to explore the skin microbiome in this review.

The *Lactobacillus* example here needs more detail beyond simply talking about ‘the immune response’. Which arm of the vertebrate immune response? And what pathogen does this help to protect against?

Updated to include more detail: “For example, *Lactobacillus* in the fish gut are able to stimulate the production of inflammatory cytokines, and thus help protect against diseases through colonisation of bacterial pathogens (He et al., 2017).”

4.1 I like this section but it needs more detail given that it is about manipulation and aquaculture.

Is it sufficient to talk about genetic selection changing the composition of microbiomes? Don't we care more about microbiome function, how that is impacted by selection on host genes, and what that means for fish health? If there's no evidence of functional changes, is this an opportunity to highlight the paucity of data? You do this well with the diet section below in reference to the unknown effect of insect protein on gut microbiome function.

More detail provided here in the artificial selection section: “Additionally to host selection, artificial selection could be acting on the bacterial portion of the hologenome, as has been shown in zebrafish (Robinson et al., 2018). Bacterial communities selected by artificial selection could be neutral, sympathetic or antagonistic to the goals of the artificial selection on the host, however, and we have little understanding of this relationship, but it will be vital in manipulation of the hologenome.”

4.3 Perhaps a reference to support the statement about disease outbreak and a reliance on chemotherapeutics?

Removed.

4.4: You mention probiotics in section 4.3, which could of course be applied indirectly through the environment, so should probiotics alternatively be discussed in this section?

Very good point! Addition of: “Bacteriophages and probiotics have also been applied directly to the directly to tank water, with promising results (Moriarty, 1998; Skjermo and Vadstein, 1999; Almeida et al., 2019) (figure 4b), although with certain common RAS equipment removed, such as ultraviolet water sterilisation units. Far less is known about this method of application, however, when compared to adding probiotics to feed (Jahangiri and Esteban, 2018), and provides potential for future research.”

4.4: “allow for a microbial...” something missing here

Fixed.

Figure 3: I like what you're trying to do here but I think it could be clearer. For example, section (a) should really include the effect of feed as prebiotics on the microbiome, but seems to only deal with direct microbial input. Section b seems odd when presented as 3 overlapping discs. I'd suggest it would be more informative to open this up to allow illustration of how the three discs are inter-related using arrows. For example, the abiotic gut env. Can affect both microbe-microbe and host-microbe interactions. Finally, section c is an opportunity to illustrate how we can use high throughput techniques to measure both the structural and functional characteristics of teleost microbiomes, but critically that these two traits a) tell us different things and b) do not always map 1:1 because of horizontal gene transfer, functional redundancy across microbial species, and the importance of understanding gene expression and not just gene presence in microbial genomes.

Some great feedback here, thank you, which I have utilised in expanding figure 3 into a far more informative diagram.

Appendix C

Associate Editor Board Member

Thank you for your constructive feedback, it is very much appreciated. Please find my responses, and changes that I have made within the manuscript to try and address the weaknesses highlighted. Some of the quotes below may also have been reworded in the final edit in order to minimise word count and keep to the 10-page limit required by Proceedings of the Royal Society B.

Comments to Author:

I agree with the reviewers that the manuscript is much improved but requires further minor revisions. The reviewers highlight that further edits are needed to:

1. Better synthesise the disparate sections of the manuscript

I agree this critique. A concerted effort has been made to reference other sections. Examples of this are provided below, responding to reviewers' comments. In particular, please see comments relating to:

“If I had one criticism of the revision it's that it still feels like a suite of disparate sections stuck together in one paper, rather than a contiguous synthesis of microbiome aquaculture relationships. For example multiple sections mention *Lactobacillus*, first as immune modulators and then as probiotics, but these sections don't flow into, or reference, one another.”

2. Ensure that the immunity section is less general and more focussed on aquaculture.

This section has since been revised, with more emphasis on the application to aquaculture in the examples stated.

3. Discuss the role of the environmental reservoir of bacteria in the section that considers artificial selection.

I have now added greater emphasis to environmental bacteria: “Artificial selection on the environmental reservoir of bacteria surrounding the fish generates the potential for improving colonisation success of commensal bacteria. Currently, bacterial communities selected by host artificial selection could be neutral, sympathetic or antagonistic to the goals of breeding programs, and understanding this relationship will be vital in manipulating the hologenome.”

4. Make the connections between dysbiosis, welfare and rearing conditions more explicit.

I have tried to make the use of dysbiosis in welfare more explicit, with the addition of: “Stress biomarkers such as dysbiosis allow for optimisation of husbandry, alerting operators to chemical stressors (i.e. poor water quality, pollution, diet composition imbalance, accumulation of nitrogenous and other metabolic wastes), biological stressors (i.e. overcrowding, social dominance, pathogens), physical stressors (i.e. temperature, light, sounds, dissolved gases) or procedural stressors (i.e. handling, transportation, grading, disease treatment) (Gabriel, Gabriel and Akinrotimi, 2011).”

5. Make it clearer exactly how filling in specific missing pieces of data will advance the efficacy of aquaculture.

On reading the manuscript, I also have some further suggestions:

Considering that the review is focussed on the gut microbiome of teleost fish, it is curious that other than sporadic references to specific taxa like *Lactobacillus* and *Bacteroides*, the manuscript does not provide a textual or graphical description of the taxonomic composition of the teleost gut microbiome, or the 'pathogens' that are referred to. Providing such information could assist the authors in addressing some of the points raised by the reviewers above, as they can specifically describe associations of specific taxa with diet, immunity, health and disease, and use this to illustrate some of the conceptual points made with empirical evidence. For example, which 'beneficial' taxa would be the best candidates as probiotics? And which deleterious microorganisms would bacteriophage applications be used to target? This in turn will be of benefit to the reader, who can use this information to interpret their own teleost microbiome datasets.

I agree with this comment and have gone through the manuscript adding the names of bacterial taxa in studies we have referenced. I have avoided going into too much detail about which are beneficial, and which are deleterious, as these can be species specific. The benefits of certain bacterial taxa also depend on the ecology of the host, and differ between fish that live in freshwater/saltwater, or fish that are herbivorous/carnivorous. But you are right, there are ubiquitous taxa that are important to aquaculture, be it beneficial such as the *Lactobacillus*, or pathogenic, such as the *Vibrio* and *Aeromonas*, and I have made every effort to include these in the new revision.

Examples:

“The importance of controlling pathogens such as *Vibrio* sp. and *Aeromonas* sp. will also be discussed in relation to the fish gut microbiome and closed loop systems later in the review.”

“Currently, gut bacterial communities have been assessed in over 145 species of teleost fish from 111 genera, representing a diverse range of physiology and ecology (figure 1), often with similarities in bacterial pyhla composition between fish species, dominated by Bacteroidetes and Firmicutes (Sullam et al., 2012; Givens et al., 2015).”

“For example, the amount of vitamin B12 positively correlated with the abundance of anaerobic bacteria, belonging to the genus *Bacteroides* and *Clostridium*, in Nile tilapia (*Oreochromis niloticus*) (Sugita, Miyajima and Deguchi, 1990).”

“As insects are rich in chitin, these diets have been associated with prebiotic effects, through increased representation of beneficial commensal bacteria, genus such as *Pseudomonas* and *Lactobacillus*, which in turn, improves performance and health in some fish hosts (Bruni et al., 2018).”

“Gut microbial communities of the Asian seabass (*Lates calcarifer*), for example, shifted markedly in response to an 8-day starvation period, causing enrichment of the phylum Bacteroidetes, but reduction of Betaproteobacteria, resulting in transcriptional changes in both host and microbial genes (Xia et al., 2014).”

“For example, *Bacillus* and *Lactobacillus*, two common probiotics used in aquaculture, are able to stimulate expression of inflammatory cytokines in the fish gut (He et al., 2017)”

The authors address four key themes in the paper : 1) diet, 2) immunity, 3) artificial selection, 4) removing aquaculture from ecosystems (closed loop systems). But as they also numbered the sections in the manuscript, which means that themes 1, 2, 3 and 4, which follow the introduction, are numbered as sections 2, 3, 4 and 5 in the manuscript, which is a little jarring. I would suggest removing these section numbers to avoid this issue.

Numbers removed

The manuscript needs a careful proofread for missing words, to correct variable spacing between words (e.g. after 'halibut' in line 129 and before 'immunity' in line 144), and typos.

Fixed

Line 184 – the full genus name is needed for *A. hydrophila*

Fixed

Reviewer(s)' Comments to Author:

Referee: 2

Many thanks for your constructive feedback, I have tried my best to amend the weakness highlighted.

Comments to the Author(s).

The authors have largely addressed the reviewers' comments in the revision.

The Immunity section was not related to aquaculture. I think this should be done since the objective of the revision is to review studies of the microbiome as related to aquaculture.

I agree. This section has since been revised, with more emphasis on the application to aquaculture in the examples stated.

The Artificial Selection section includes the concept of the hologenome. This can be opening a can of worms. At least the role of the environmental reservoir of bacteria should be mentioned.

I have now added greater emphasis to environmental bacteria: “Artificial selection on the environmental reservoir of bacteria surrounding the fish generates the potential for improving colonisation success of commensal bacteria. Currently, bacterial communities selected by host artificial selection could be neutral, sympathetic or antagonistic to the goals of breeding programs, and understanding this relationship will be vital in manipulating the hologenome.”

In section 5 (iii), The authors can make the connections between dysbiosis, welfare and rearing conditions more explicit. Presumably the idea is to change rearing conditions to minimize stress, maximize growth, etc.

I have tried to better contextualise the use of dysbiosis in welfare, with the addition of: “Stress biomarkers such as dysbiosis allow for optimisation of husbandry, alerting operators to chemical stressors (i.e. poor water quality, pollution, diet composition imbalance, accumulation of nitrogenous and other metabolic wastes), biological stressors (i.e. overcrowding, social dominance, pathogens), physical stressors (i.e. temperature, light, sounds, dissolved gases) or procedural stressors (i.e. handling, transportation, grading, disease treatment) (Gabriel, Gabriel and Akinrotimi, 2011).”

In the Conclusions section, the authors could make it clearer exactly how filling in specific missing pieces of data will advance the efficacy of aquaculture. If it makes sense to do so, the authors can also prioritize the types of data to collect or to make it clear that it is necessary to collect some types of data before it makes sense to collect other types of data.

I have included RNA sequencing, metabarcoding and metagenomics, fluorescence in situ hybridization and population genomics in brackets in the conclusion section, in order to emphasize the types of data needed to be collected without sounding too repetitive.

Figure 3 is not fully described in the text.

Fixed

70: Can the authors tie specific interactions to specific references?

Fixed: “More in-depth reviews focusing on specific interactions with teleost hosts are available, for example, interactions between the gut microbiome and the immune system (Kelly and Salinas, 2017), energy homeostasis (Butt and Volkoff, 2019), physiology (Yukgehnaish et al., 2020) and the core gut microbiome (Tarnecki et al., 2017)”

148: Cytokine production is a double-edge sword, but I take the authors’ point.

150: The reference to bacteria-specific gene expression lacks context. What genes are expressed and to what end?

More details added: “For example, *Bacillus* and *Lactobacillus*, two of the most commonly used genus of bacteria for probiotics in aquaculture, are able to stimulate expression of inflammatory cytokines in the fish gut (He et al., 2017), increase the number of goblet cells (responsible for the production of a protective mucus layer) (Popovic et al., 2017), and increase phagocytic activity, among other innate immune responses (Chen, Liu and Hu, 2019).”

186: What is meant by “direct opposed response”?

Clarified: “(Liu et al., 2015). Results from their study suggest that oral vaccines can target specific genera (e.g. *Aeromonas*) through activation of innate and adaptive immune defences within the intestine without causing large disturbances in non-target microbiota populations.”

222: What is meant by “proteins encoding for immune parameters”?

Clarified: “Several prebiotics have been reported to manipulate the resident microbiota community of a host in favour of Firmicutes and short-chain fatty acid producing communities (Piazzon et al., 2017), while increasing levels of antimicrobial enzymes such as myeloperoxidase (Xu et al., 2018) and enhancing activity of innate defences (Geraylou et al., 2013).”

229: presumably genetic components of the host?

Clarified: “Recent evidence suggests host genetics may play a fundamental role in determining the gut microbiota community in fish (Li et al., 2018).”

324: It seems that what are lacking are not elements of experimental design but lack of response variables.

Changed: “As highlighted in figure 1b, there are still key elements lacking from many microbiome studies, particularly those assessing meta community composition, with the lack of water samples being particularly glaring.”

340: Please define “mechanisms that underly [sic] function”.

Clarified: “Furthermore, a large knowledge gap exists between the composition of the microbiome and its functional expression, this is partly due to the complexity and variability in the ecology of teleost gastrointestinal tracts (Egerton et al., 2018) and unknown bacterial taxa.”

Carefully check the manuscript for typos and other errors, e.g., lines 92 (“examplem”), 139 (“all fish in a are”) and 152 (micrboes).

Fixed – and apologies!

Referee: 3

Thank you for your constructive comments up to this point, and your part in making the manuscript a better quality!

Comments to the Author(s).
Review of Perry et al ProcB Revision

This manuscript has had quite an overhaul since the previous round of review. I think it does a far better job of satisfying the brief / matching the title now. I find the figures to be improved, especially Fig. 3.

Thank you!

If I had one criticism of the revision it’s that it still feels like a suite of disparate sections stuck together in one paper, rather than a contiguous synthesis of microbiome aquaculture relationships. For example multiple sections mention *Lactobacillus*, first as immune modulators and then as probiotics, but these sections don’t flow into, or reference, one another.

I agree. I have tried to tie the sections together, using specific taxa such as *Lactobacillus*, as suggested, to link sections.

Examples:

Immunity

“The importance of controlling pathogens such as *Vibrio* sp. and *Aeromonas* sp. will also be discussed in relation to the fish gut microbiome and closed loop systems later in the review.”

(ii) Pro- and prebiotic supplementation

“As has already been discussed, probiotic bacteria belonging to the genus *Bacillus* and *Lactobacillus* have a beneficial effect on immunity and are suggested to provide an alternative approach to controlling disease in aquaculture.”

(i) Manipulating environmental microbiota section

“Successful application of RAS and BFT could therefore be combined with vaccines for bacteria such as *Aeromonas* sp., as previously discussed, to reduce infections.”

“Supplementation of live feed cultures with beneficial microbes, such as the previously mentioned *Lactobacillus* sp.,”

Secondly, the MS mentions RAS aquaculture in the first few paragraphs but doesn't then use it to reference the entire dedicated section on RAS at the end of the manuscript where more, vital detail can be found. This means it feels like these sections could have appeared in any order in the paper without affecting the structure, and also then sounds a bit repetitive, which seems jarring. I appreciate the authors have a lot of ground to cover here, and so I think my criticism is an emergent property of that. But I think there is still room for improvement here.

Linking sentence added:

“Even if all fish are terminated shortly after starvation, gut microbial community changes before termination could cause long term impacts to the microbial composition of the water and biofilters in closed recirculating aquaculture systems (RAS) (figure 4b). RAS systems will be discussed in greater detail later in this review.”

Below I list some issues where I think further clarity is needed as there are quite a few places throughout the MS where the language feels a bit imprecise. I also found a couple of typos.

Line 53: I think it would be good to expand the description of these examples to see how the gut microbiome is relevant. For example, is the clownfish example looking at the influence of feeding rate on the gut microbiome?

Clarified: “. Examples of studies on non-model teleost gut microbiomes range from those demonstrating rapid gut microbiome restructuring after feeding in clownfish (Parris, Morgan and Stewart, 2019) to the effect of differing environmental conditions, such as dissolved oxygen content, on the gut microbial diversity of blind cave fish (Ornelas-García et al., 2018).”

Line 92 “example”. I'd also suggest you split the sentence here into two separate ones to avoid the excess of commas

Clarified: “For example, the amount of vitamin B12 positively correlated with the abundance of anaerobic bacteria belonging to the genus *Bacteroides* and *Clostridium*, in Nile tilapia (*Oreochromis niloticus*)(Sugita, Miyajima and Deguchi, 1990).”

Line 98 is there an ‘and’ missing after ‘high quality protein’?

Clarified: “Fishmeal is an efficient energy source containing high quality protein, as well as highly digestible essential amino and fatty acids (Cho and Kim, 2011).”

Line 111: I think you should state that these antinutrients are found in the soy meal as it's not clear what these relate to in the current sentence

Clarified: “Plant-protein sources have been shown to disturb the gut microbiota of some fish, with the production of antinutritional factors (factors that reduce the availability of nutrients) and antigens, impeding host resilience to stress (Batista et al., 2016), metabolism (Gatesoupe et al., 2018) and immune functioning (Miao et al., 2018).”

Line 114: ‘host function’ is a bit vague here

Removed

Line 118: are there references for the statement about altered microbial diversity?

Addition of (Desai et al., 2012; Miao et al., 2018).

Line 122: is this because chitin is a prebiotic? Might be worth spelling out for a non-specialist reader

Clarified: “As insects are rich in chitin, these diets have been associated with prebiotic effects, through increased representation of beneficial commensal bacteria, genus such as *Pseudomonas* and *Lactobacillus*, which in turn, improves performance and health in some fish hosts (Bruni et al., 2018).”

It has prebiotic effects in some fish, but as explained later in the paragraph, not all fish react the same, and so cannot be considered to be a prebiotic all of the time.

Line 141: I like this section on starvation, but could you expand on why starvation-mediated microbiome changes might be important in RAS aquaculture? I don't follow it.

Clarified: “Even if all fish are terminated shortly after starvation, gut microbial community changes before termination could cause long term impacts to the microbial composition of the water and biofilters in closed recirculating aquaculture systems (RAS) (figure 4b). RAS systems will be discussed in greater detail later in this review.”

Line 144: Too many commas in this sentence for my liking as I think It could flow better

One comma removed

Line 146: What's a robust gut microbiome? How do we measure it ?

Amended: “Therefore, a microbially diverse gut microbiome in aquaculture is important to prevent unfavourable microbial colonisation (Balcázar, Decamp, et al., 2006)”

Line 152 “microbes”

Fixed

Line 161: Is the wild stickleback causal chain study missing a reference? I think it's this study Friberg IM, Taylor JD and Jackson JA (2019) Diet in the Driving Seat: Natural Diet-Immunity-Microbiome Interactions in Wild Fish. *Front. Immunol.* 10:243

It was! Apologies!

Line 286: I know you reference fig 4b, but what are the promising results here?

Clarified: “Bacteriophages and probiotics have also been applied directly to tank water (figure 4b), with probiotics such as *Bacillus* sp. preventing loss from *Vibrio* sp. infections (Moriarty, 1998), and *Flavobacterium columnare* -infecting phages have been shown to persist in RAS for up to 21 days (Almeida et al., 2019);”

Figure 3: Much improved from V1 of the manuscript, but I can't see the a-d subsections as indicated in the legend that I assume relate to the bottom right corner. Also where is 3c?

Fixed

Appendix D

Many thanks again for your comments and constructive feedback!

Line 150 - 'For example, Bacillus and Lactobacillus, two of the most commonly used genus ' genus (singular) should be changed to 'genera' (plural).

Fixed.

Also here: 'As has already been discussed, probiotic bacteria belonging to the genus Bacillus and Lactobacillus'

Fixed.

And here: 'beneficial commensal bacteria, genus such as Pseudomonas and Lactobacillus, which in turn,'

Fixed.

Lines 155 and 267 - 'Vibro' should be Vibrio?

Fixed.

Line 226 - 228 - the term 'Artificial selection on the environmental reservoir of bacteria surrounding the fish' should be revised as 'artificial selection' cannot act directly upon the environmental reservoir of bacteria. Maybe 'environmental filtering' is a more appropriate term.

Fixed.

"Supplementation of live feed cultures with beneficial microbes, such as the previously mentioned Lactobacillus sp.," One species? If more than one species, need to edit to spp.. Also here; 'Bacillus sp. preventing loss from Vibro sp. '

Fixed.

There seems to be a word missing in this sentence: 'and Flavobacterium column are -infecting phages'

Fixed.